# Tooth Enamel and Its Dynamic Protein Matrix

**DOI:** 10.3390/ijms21124458

**Published:** 2020-06-23

**Authors:** Ana Gil-Bona, Felicitas B. Bidlack

**Affiliations:** 1The Forsyth Institute, Cambridge, MA 02142, USA; 2Department of Developmental Biology, Harvard School of Dental Medicine, Boston, MA 02115, USA

**Keywords:** tooth enamel, enamel proteome, amelogenin, amelogenin-Y (AMELY), enamel peptide, molar hypomineralization, dental anthropology, dental fluorosis, serum albumin

## Abstract

Tooth enamel is the outer covering of tooth crowns, the hardest material in the mammalian body, yet fracture resistant. The extremely high content of 95 wt% calcium phosphate in healthy adult teeth is achieved through mineralization of a proteinaceous matrix that changes in abundance and composition. Enamel-specific proteins and proteases are known to be critical for proper enamel formation. Recent proteomics analyses revealed many other proteins with their roles in enamel formation yet to be unraveled. Although the exact protein composition of healthy tooth enamel is still unknown, it is apparent that compromised enamel deviates in amount and composition of its organic material. Why these differences affect both the mineralization process before tooth eruption and the properties of erupted teeth will become apparent as proteomics protocols are adjusted to the variability between species, tooth size, sample size and ephemeral organic content of forming teeth. This review summarizes the current knowledge and published proteomics data of healthy and diseased tooth enamel, including advancements in forensic applications and disease models in animals. A summary and discussion of the status quo highlights how recent proteomics findings advance our understating of the complexity and temporal changes of extracellular matrix composition during tooth enamel formation.

## 1. Introduction

In recent years, the number of proteins implicated in tooth formation and, in particular, tooth enamel mineralization has grown considerably. This is partly due to studies on gene expression, partly based on advances in proteomic analyses of tooth enamel that have provided an expanded list of proteins and peptides. While these findings deepen our understanding, they also highlight the unique features of the mineralizing tooth enamel matrix, such as proteins specific to tooth enamel. One key difference between tooth enamel, bone and dentin is that the protein matrix changes during enamel formation dramatically in both abundance and composition, decreasing from 35 percent dry weight leaving in erupted teeth less than one percent by weight of protein as an entombed fossil record of ontogeny [1]. 

Tooth analyses are increasingly used in the biomedical field to obtain individual histories of health and exposure to adversities during prenatal as well as postnatal development [2,3]. This use of teeth as a biomarker leverages several unique features of tooth formation, including the known timing of tooth development in humans during discrete periods in ontogeny, an incremental process of formation, the absence of turn-over after formation and the accessibility of shed or extracted teeth [4]. However, there is growing interest in unlocking the organic record preserved in tooth enamel. This is a prerequisite to better understand the factors contributing to enamel formation but also gives access to histories of exposure to organic toxicants. Proteomic analyses have become critical to expand our view of the mineralizing enamel matrix and to decipher the information the organic phase holds. It seems therefore timely to review and discuss proteomic analyses of enamel considering our current understanding of healthy and diseased enamel, disease models, as well as forensic applications. 

## 2. Tooth Enamel Formation

Our teeth are composed of a hard layer of tooth enamel over a softer dentin core. Cementum covers the tooth root and anchors the periodontal ligaments that attach the tooth to the jaw. The tooth dentin is innervated and the tooth connected to vascularization through the dental pulp, located in the pulp cavity. Tooth enamel is extremely hard, in fact, the hardest substance in our body, without being brittle. It is a composite material with outstanding properties, designed to last a lifetime withstanding chewing forces and extreme chemical and temperature variations, as well as protecting the tooth structure from external damage. The formation of tooth enamel, amelogenesis, takes place inside the jaw and before tooth eruption. Because this makes it difficult to obtain tooth samples during early stages enamel formation, the rodent model is especially helpful with continuously forming incisors that allow for the study of all stages of enamel formation within an incisor. Tooth enamel mineralization starts with orchestrated signaling pathways regulating the differentiation and well-timed activity of specialized cells [5]. The enamel forming cells, ameloblasts, secrete an extracellular matrix where unique enamel proteins and proteases regulate shape, arrangement and mineralization of calcium phosphate crystals [6]. 

The formation of tooth enamel is typically described as a sequence of consecutive stages that starts with presecretory, then secretory, through the short transition, to maturation stage, followed by apoptosis of ameloblasts and tooth eruption into the oral cavity (Figure 1) [1]. After ameloblasts differentiate during the presecretory stage, they enter the secretory stage with their morphology characterized by an oblique cell process, the Tomes’ process. Secretion of the mineralizing enamel matrix begins at the interface with dentin creating a junction between two different mineralized materials with different properties and formation processes, the dentin-enamel-junction (DEJ). Deposition of the mineralizing enamel matrix occurs with a circadian rhythm leaving a track of daily increments during the secretory stage until full enamel thickness is reached and ameloblasts differentiate further. During this stage, enamel is still soft due to the high amount of enamel matrix proteins and has a cheese-like consistency allowing for easy sample collection by scraping off. After a short transition stage, ameloblasts change their morphology. The Tomes’ process, widely held to be critical for anisotropic matrix deposition resulting in bundles of enamel crystallites (rods), disappears. Ameloblasts alter their apical cell morphology and enter a cyclical transition between smooth and ruffle-ended. These alternating morphologies are a hallmark of maturation stage and parallel fluctuation in both enamel matrix pH and ion transport that accompany the massive crystal growth. The changes in cell morphology also reflects the change in activity from secretion to uptake of cleaved enamel matrix proteins to allow for the expansion of mineral crystals. As a result, protein abundance decreases while the mineral content increases dramatically. The enamel crystals mature and grow until they interlock within bundles in the completed enamel, where mineral constitutes 95 percent by weight and only about 1 to three percent by weight of protein remains [7,8] (Figure 1). It is well established that the structural enamel matrix proteins as well as proper ion transport and pH regulation are indispensable for correct enamel mineralization [6,9,10]. 

The last decade has seen numerous technological and methodological advances in proteomic analyses, resulting in important improvements of accuracy, sensitivity, speed and throughput of mass spectrometry analyses [13]. Based on these advances, researchers have uncovered new functions and roles of proteins in enamel by their identification, location, quantification, analysis of post-translational modifications and protein-protein interactions [14]. With an increasing number of enamel matrix components being uncovered, the following questions arise—which proteins are endogenous, which ones exogenous, which ones are essential, how do they contribute to enamel development and when do they play a role for enamel formation as well as function? [15]. The answers to these questions are key to developing clinical benefits, such as improved diagnosis and repair, to leverage biomimetic approaches to regenerate enamel or to create new materials with the exceptional properties of healthy tooth enamel [16,17,18]. For the use of teeth as biomarkers, it is critical to learn more about what information this fossilized protein record holds for us to decipher about ontogeny, behavior, adverse exposures and the environment [3]. 

## 3. Teeth as Records: Enamel Proteomics in Archeology, Forensics and Odontology

Tooth enamel has been used as a record of ontogeny, health history and as a proxy record for behavior and environment in archeological research and for forensic purposes. The degradation-resistant proteins remain well preserved within the densely packed and mineralized tissues, including the highly mineralized tooth enamel [19,20,21,22,23,24]. Proteins located within highly mineralized remains, such as tooth enamel, appear to be well preserved for millions of years and withstand postmortem diagenetic alteration better compared to proteins in more porous mineralized tissues such as bone. For example, the proteomic analysis of dental enamel has allowed addressing the phylogenic relationship of the Eurasian Rhinocerotidae and additional information from other fauna from 1.77-Myr old tooth [25]. This is because the mineral-bound protein is less likely to be lost and located between densely packed mineral crystals remains well preserved for longer duration than nucleic acids [26,27,28]. The study of ancient proteins preserved in fossils is named paleoproteomics. It is complementary to genetic and morphological studies and has gained impact with advances in mass spectrometry as reflected in the number of available techniques and aspects of analyses such as functionality, sensitivity, resolution and type of data produced [29]. 

The preservation differences between tissues and bone and teeth are important for the identification of sex in human remains, which is important in forensic sciences, archaeology and paleoanthropology. Classical approaches of sex determination are based on canonical osteological methods and DNA analysis in bone [30,31,32,33]. The analysis of the enamel protein amelogenin is used to determine the sex of an individual, when the preservation of bone and DNA is compromised, or contaminations prevent reliable analyses [34]. Amelogenin is encoded by two genes, *AMELX* gene on the X chromosome X and *AMELY* on the Y chromosome in several mammalian families, including hominidae, suids and bovidae. In contrast, rodent species, have only one *AMELX* [35,36,37,38,39,40]. Variations in the genomic sequences of amelogenin encoded from the X and Y chromosomes allow the identification in forensic investigation of female, isoform X or male, isoforms X and Y, in human enamel [41,42]. 

The power of enamel proteins and proteomics as a robust method for sex identification is demonstrated in the case of the “Lovers of Modena,” in which the determination of sex allows us to see history under a different perspective [43]. In the cemetery of Modena (Italy), the burial site of two adults holding hands was uncovered and dated to the 4th-6th century. Based on the intimate gesture of their position in the excavation, the skeletons were held to belong to a man and a woman in love. Due to the poor preservation of the skeletons the sex remained undetermined. However, with extraction of sample material for proteomic analyses, the AMELY isoform has been identified in the enamel of the two lovers (Table 1). It turned out that both adults were male. This study also included analyses from other skeletons and modern teeth in which other peptides from AMELX, AMELY and enamelin were identified (Table 1 and Table 2). The method used for sex identification is similar to the procedure published by Stewart et al. [44] but different protein extraction methods were used (Figure 2). The procedure used by Stewart et al. is based on determination of sex identifying AMELY-(58–64) peptide (SMoxIRPPY), found only in male samples and AMELX-(44–52) peptide (SIRPPYPSY) identified in both, males and females (Table 1). 

Other innovative studies have used amelogenin peptides for sex determination, for example, in fossils up to 7,300 years old using either enamel or the whole teeth for the analyses [51,52]. These studies have tried to improve the peptide extraction and identification to avoid false female assignment. The problem of “female by default” may be due to the lower expression of AMELY compared to AMELX. The *AMELY* gene expression is only 10% of the *AMELX*, hampering the AMELY protein identification [40]. The effect should be exacerbated in whole tooth analyses, where more abundant proteins are much more predominant.

The refinement of methods for sample extraction and analysis is key to addressing a wide range of fascinating questions where the composition of the extracellular matrix holds the information of interest in either fossil or modern material. Castiblanco et al. [45] investigated the enamel protein composition from healthy contemporary teeth and pulverized the enamel to extract the proteins. These authors used liquid chromatography coupled to tandem mass spectrometry (LC-MS/MS) analysis without previous trypsin digestion (Figure 2). Enamel peptides were successfully identified, although only the low-molecular-weight ones (Table 1 and Table 2). Seven proteins from human erupted third molars were identified, including proteins specific for human enamel (amelogenin isoforms X and Y, enamelin and ameloblastin) and non-specific proteins (serum albumin and antithrombin). In other similar work, specific enamel proteins and several non-tooth specific proteins were identified from mature human extracted teeth using nano LC-MS/MS (Table 1 and Table 2) [46]. In situations of limited sample quantity, the high analytical sensitivity of nano LC-MS/MS is especially advantageous for detection and identification [53]. For this analytical approach, Stewart et al. [46] used a variation of the acid etch method for protein extraction and compared between sampling methods with and without trypsin digestion (Figure 2). Although the trypsin digestion increased the variety of peptides, no big differences were observed between the two methods for enamel specific proteins (Table 1 and Table 2). Amelogenin (X and Y isoforms), enamelin, ameloblastin and serum albumin were identified by both methods (with and without trypsin) (Table 2). However, collagen and alpha-1-antitrypsin were only identified by trypsin digestion process (Table 2). Enamel proteins were also identified in this study from archeological enamel samples without trypsin digestion (Table 1 and Table 2).

Recently, the oldest enamel proteomes were obtained from Early and Middle Pleistocene hominid teeth [54]. The drilled off enamel, potentially with some traces of dentin, was subjected to digestion-free peptide extraction and analysis by nano LC-MS/MS. The authors found that the average peptide lengths decreased with age of the enamel sample, yet, in vivo modifications were still present. This study identified enamel-specific proteins, including enamelin, ameloblastin and Mmp20, as well as AMELX and AMELY versions of amelogenin, in addition to serum albumin and collagens. The application of this methodology to contemporary and archeological enamel samples yielded some differences in protein identification and a higher variety of peptides with the use of trypsin. These results validate the amazing persistence and preservation of enamel proteins in teeth for very long periods of time.

## 4. Enamel Protein Composition at Different Developmental Stages

An ongoing effort in enamel research is to identify all enamel matrix proteins in the different stages of enamel development and understand their functionality. All three of the classic enamel proteins, amelogenin, ameloblastin and enamelin are known to be essential to form healthy enamel. In addition, matrix metalloproteinase-20 (Mmp20) and kallikrein-4 (Klk4) are two enzymes that are required for proper enamel formation and their activity of sequential proteolysis of enamel proteins is well studied [1]. These two resident proteases facilitate the progressive removal of matrix proteins, thus providing space for apatite crystals to grow and generate an enamel layer that is harder and less porous [55]. Mmp20 is present and active from secretory to maturation stage. The proteolytic cleavage of amelogenin by Mmp20 prevents aberrant mineral deposition during the secretory stage [56]. Ameloblasts start to secrete Klk4 during the transition from secretory to the maturation stage. Klk4 cleaves previously secreted matrix proteins, including the ones partially hydrolyzed by Mmp20 and is critical for the completion of enamel mineralization and to reach full enamel hardness. Despite their expression being independent of each other, Klk4 activation is regulated by Mmp20 and Mmp20 inactivation is regulated by Klk4 [57]. The absence of Klk4 does not affect enamel thickness or basic microstructure, that is the arrangement of crystal bundles [58].

## 5. Structural Enamel Matrix Proteins

Of the three structural enamel matrix proteins, amelogenin is the most studied. This is partly due to its high abundance allowing for the extraction and purification of native amelogenin from big teeth such as from cows and pigs. In addition, amelogenin can be produced in recombinant form, albeit without N-terminal methionine and in unphosphorylated form. The secreted forms, alternative splice products and cleavage patterns of the full-length amelogenin parent protein are well known [1]. Upon its secretion, the full-length amelogenin is short lived and found only in proximity to the ameloblasts [59] before its cleavage sequence is initiated by Mmp20 to produce a suite of amelogenin proteins that vary in length and abundance and occur with amelogenin isoforms that result from alternative splicing of RNA [36,40,60,61,62,63,64,65,66]. Although the N- and hydrophilic C-terminal region of amelogenin is highly conserved, some variability in the hydrophobic center is more variable [67]. As a consequence, the number of amino acids of the full-length amelogenin varies slightly between different species and is 189 in humans, 180 in mice and 173 in pigs. In contrast to many other proteins known to regulate biological mineralization, amelogenin as well as its alternative splice products, have only one phosphorylation site at serine-16 [67,68]. One needs to keep in mind though, that amelogenin is a relatively small molecule, yet, highly concentrated in the extracellular matrix. It has been shown that this single phosphorylation on serine-16 is of critical importance for mineralization and plays a key role for the regulation of mineral phase and transition from amorphous calcium phosphate to hydroxyapatite [69,70,71,72,73]. The most abundant alternative splice product of amelogenin is the Leucine Rich Amelogenin Peptide (LRAP), with proposed roles in enamel formation, as cell signaling molecule or in the regulation of crystal morphology [67,74,75,76,77,78,79]. Recent studies have provided evidence that the phosphorylation of LRAP induces structural changes that in turn affect enamel mineral formation [69,80]. Whether and when amelogenin is dephosphorylated during amelogenesis is not well known but the question has been addressed in our previous proteomic analyses [15].

The amelogenin cleavage sites and sequence of digestion through Mmp20 activity starting with the secretory stage and Klk4 activity starting in transition stage is well documented [36,81]. The final proteolytic peptide of amelogenin is the tyrosine-rich amelogenin peptide (TRAP) which is accumulated in the small amount of protein matrix left in mature enamel [36,82,83,84]. Despite the low protein content in erupted teeth, the amelogenin X isoform of TRAP was identified by matrix-laser desorption/ionization time-of-flight/time-of-flight mass spectrometry (MALDI-TOF/TOF) from skeletons (Neanderthal and medieval) and contemporary teeth after removing the enamel by drilling followed by trypsin digestion (Figure 2) [49]. Another study using the same methodology but collecting only proteins smaller than 30 kDa, characterized several amelogenin peptides from contemporary as well as mummy molars (Table 1 and Table 2) [21]. What improved the enamel protein extraction in the Porto et al. study [21], is the method of superficial enamel etching and the analysis if proteins smaller than 30 kDa (Figure 2). This allowed for the identification of small amelogenin peptides that could be eclipsed by other larger and more abundant proteins and opened new perspectives for enamel protein identification. The drawback of this method is that large proteins or peptides could be missed.

The proteomes of different parts of the tooth crown, namely tooth enamel, dentin and DEJ and the adjacent enamel organic matrix (EOM) (DEJ + EOM) proteomes were recently separately analyzed [47]. Laser capture microdissection was applied to sectioned healthy erupted human molars to sample the different layers followed by protein extraction and nano-LC-MS/MS analysis (Figure 2). The 49 reported proteins found in enamel included ameloblastin, collagen and serum albumin (Table 2). Whereas isoform 3 and 1 of *AMELX* were not detected in enamel or dentin but seen in the DEJ + EOM sample. It is surprising that no amelogenin peptide findings were shown for enamel, while isoform 2 of ameloblastin was identified in enamel and DEJ + EOM and not in dentin. Some proteins, including several collagen isoforms and serum albumin, were detected in all four tooth compartments of dentin, enamel and DEJ + EOM. The authors also reported several members of the serpin family, including antithrombin-III and alpha-1-antitrypsin. These proteins were also found in dentin but not in DEJ + EOM. Despite successfully addressing the need to separate the different components of forming teeth in proteome analyses, the study also implies that tooth proteome analyses are still challenging, since most of the enamel-specific proteins were not identified.

There are several hurdles for the analysis of human dental samples. One is the difficulty to obtain human samples under conditions that are standardized, reproducible and conducive to proteomic analyses, although these conditions are required to establish sample homogeneity. Another hurdle is that there might be tradeoffs between precision in sampling methods and accuracy in protein isolation that are of key importance to avoid loss of proteins, contaminations, introduction of false positives and compromised integrity of proteins and data. It is encouraging though, that proteome analyses of human fossil material demonstrate the feasibility to obtain data on protein composition in fossil human enamel [43,44,46,54].

## 6. Enamel Matrix Removal and Proteins Remaining in Erupted Tooth Enamel

A small amount of protein remains in tooth enamel after mineralization is completed and teeth are erupted. Albeit small, the amount is critical and deviations in protein content diminish enamel integrity and function [85]. Given the high content of amelogenin in developing, pre-eruptive enamel, it is not surprising to find amelogenin proteins or peptides in the erupted tooth crown. The rates of degradation are not well characterized for all enamel proteins. Therefore, it is not clear whether non-amelogenin proteins remain intact or are degraded at same rate as amelogenin or whether and how these processes are affected by post-translational modifications, such as glycosylation. The importance of these questions becomes immediately apparent when the process of protein removal is altered. Both the timing and the extent of protein removal affect the composition of the organic matrix. As a consequence, protein-mineral interactions change in forming enamel and regulate structural organization, mineral phase and composition, as well as crystal growth. The rate of protein degradation is also relevant if teeth are used as proxy-records and data from the enamel proteome leveraged as a biomarker.

Compromised enamel and dental defects can arise when the enzymatic degradation or removal of proteins during enamel formation is dysregulated. For example, amelogenesis imperfecta (AI) describes hereditary disorders of compromised enamel functionality that comprise a variety of different clinical presentations, dependent on the gene or genes affected [86]. Three different variations of AI have been described—(1) hypoplastic AI results in thin enamel and is caused by secretory stage failure; (2) in hypomaturation AI full enamel thickness is reached but the enamel is softer due to the incomplete removal of proteins during the maturation stage; and (3) hypocalcified AI, characterized by insufficient transport of calcium ions into the enamel and, hence diminished crystal growth. The ever expanding list of human gene mutations includes genes encoding enamel proteins and the proteases *MMP20* and *KLK4* [86,87]. Previous works showed that hypocalcified and hypomaturation AI enamel contains 2 to 5 percent by weight of protein. This is a two to 50-fold increase in protein content compared with healthy enamel, where it ranges from 0.01 to 1 weight percent [1,88,89]. Amino acid analyses have indicated higher protein content in hypocalcified and hypomaturation AI teeth compared to healthy enamel [90]. In contrast, the protein content in hypoplastic AI teeth was found to be similar to normal enamel (Figure 3). These results were validated later by dot-immunobinding analyses and western blot where amelogenin was detected in considerable amounts [90,91]. The studies performed in patients with AI are mostly of genetic nature and seek to relate the clinical presentation and phenotype to affected genes.

## 7. Fluoride Affects Ameloblast Function and Matrix Composition

In addition to developmental factors that regulate the highly orchestrated processes of enamel mineralization some environmental factors play a prominent role, such as fluoride. The discussion and research on the benefits and risks of fluoride for enamel formation and mineralization is ongoing. It is well-stablished that fluoride in both topical application and low dose systemic exposure increases the caries resistance of enamel because fluoridated calcium phosphate is less soluble under acid exposure [96,97,98]. However, excessive systemic fluoride exposure during tooth development results in structural and compositional changes of the enamel, dental fluorosis [89,95,99,100,101,102,103,104,105]. The protein content in fluorosed human enamel is significantly higher compared to healthy human teeth [93]. For individuals residing in areas with 3.2 ppm fluoride in drinking water, the mean protein content in enamel was 0.27 weight percent versus 0.11 weight percent in permanent human teeth (Figure 3). No mass spectrometry analysis was performed in this study, but gel electrophoresis showed fluorosed enamel with distinct bands indicating different protein patterns compared to controls.

A comprehensive proteomic characterization of proteins or peptides in mature fluorotic enamel appears not to be published yet. However, recent proteomic analyses of fluorosis-affected teeth from individuals with a 8 ppm fluoride content in their drinking water showed no different protein content compared to controls [106]. Only amelogenin peptides were identified in fluorosed teeth and control samples. The findings suggested that exposure to high concentrations of fluoride during enamel development does not affect the proteolysis of enamel proteins. More studies are necessary to understand how the protein composition in the enamel matrix is altered in dental fluorosis.

The effects of exposure to different fluoride concentrations on protein composition in enamel has been studied in the rat model at different enamel developmental stages [94]. Increased retention of amelogenins was seen in early and late maturation stage enamel compared to the control. Enamelin was also retained in the late maturation enamel at highest fluoride doses of 100 ppm. However, secretory enamel protein profiles did not show any differences with the controls. These results suggest that increased ingestion of fluoride might inhibit the mechanisms involved in protein removal rather than protein secretion. Interestingly, mice with different genetic backgrounds showed different susceptibility to dental fluorosis [107]. The A/J strain developed dental fluorosis very early under all levels of fluoride treatment, while 129P3/J mice showed minimal dental fluorosis at high doses [108]. The proteomic analyses indicated in general more proteins in maturation stage in the susceptible strain compared to the control. Opposite results were observed during secretory stage, regardless of fluoride treatment. These results validated in the rat model that in dental fluorosis enamel matrix proteins persist in mature enamel. Amelogenin, serpin and type I Collagen were exclusively identified in the maturation stage of enamel of the A/J treatment group. The presence of these proteins under fluorosis treatment may indicate either diminished protein degradation or a lack of protein inhibition processes.

To better understand the contribution of genetic backgrounds on enamel protein composition, the same group has published protein expression analyses comparing secretory and mature enamel from A/J and 129P3/J strains under regular conditions, that is without fluoride exposure [109]. Serum albumin and typical enamel proteins, such as ameloblastin, amelogenin X isoform, enamelin and Mmp20, were identified in all rat strains and conditions. Notably, Serpin H1 was exclusively detected in mature enamel from resistant mice (129P3/J), observed also in pig samples (Figure 4) [15]. Serpin H1 is a member of the serpin family of serine and cysteine protease inhibitors and plays an essential role in collagen synthesis with no inhibitory activity [110]. Serpin H1 was identified in the fluoride exposed mice, in A/J mature teeth and 129P3/J secretory teeth and the presence of type I Collagen was also observed in A/J matured enamel under treatment [108]. A recent study tried to associate the polymorphisms in enamel development genes, including type I collagen and Serpin H1, with susceptibility to dental fluorosis [111]. The polymorphisms in type I collagen and Serpin H1 did not show any difference between A/J and 129P3/J, whereas ameloblastin, type XIV collagen and Mmp20 did. Taken together, these data open a new path to validate the function and participation of these proteins in dental fluorosis.

## 8. Animal Models of Enamel Development

Animal models play a key role in dental research as access to teeth is limited during their formation inside the jaw and ethical standards, fortunately, rule out any experimental research in humans [113]. The different animal models used in dental research include rodents, canine, porcine, ovine and non-human primates. The mechanisms of enamel formation, based on involved genes and proteins, is highly conserved among mammals. The decision for a particular animal model is based on the purpose of the study, genetic tractability and similarity with human teeth [114,115,116]. Accordingly, the rodent model is routinely used and well characterized, despite differences in number and morphology of teeth [115]. The mouse model has been instrumental to study specific enamel genes and the effects of their mutation or ablation on phenotype, proteome and development, as for example Mmp20, Klk4 and amelotin null mice [56,117,118,119]. The challenge that the small size of mouse teeth poses for amount of sample material required for the analyses is often overcome by pooling samples at the developmental stage of interest. This approach was leveraged in the use of whole mouse molars. The teeth were harvested at specific and discrete postnatal time points during the first 8 days postnatally, for analyses of their protein composition by western blot and nano LC-MS/MS analysis [120]. The authors showed how the classic enamel proteins, including ameloblastin, enamelin and Mmp20 decreased one day to the next during tooth development. However, the difficulty of physically separating enamel from other tooth structures for the analysis and the use of whole teeth has an important drawback. It limits the interpretation of results pertaining to enamel development and changes in the mineralizing enamel matrix to only those matrix components that are known to be enamel specific and not expressed in dentin or pulp, since those cannot be discerned.

Pig teeth, including from miniature pig, have been used in dental research due to their availability, the large size of their teeth and the similarity in tooth size and morphology with human teeth [121,122,123]. In permanent pig teeth, interesting differences were seen in the comparison between early and late stages of enamel development [112]. The enamel samples were scraped off from unerupted extracted first permanent molars in 6-month-old pigs. The obtained samples were subjected to sequential protein extraction, fractionation and two-dimensional separation. Several glycoproteins derived from enamelin and ameloblastin were identified in the secretory-stage enamel but missing in the post secretory-stage enamel (Figure 4). The degradation of these non-amelogenin proteins correlates with proteolysis and protein removal during in the maturation stage to allow for the massive mineralization of enamel. Not all of the enamel matrix components were characterized in this study, only the most prominent proteins, including albumin. These results clearly demonstrated the degradation of a suite of proteins during the stage of enamel maturation, yet, not all enamel matrix components could be characterized. Our recent work aimed at filling this gap in knowledge and discerning changes of the mineralizing enamel matrix in different stages of enamel development and in different locations of the mineralizing tooth. The spatio-temporal pattern of protein expression in developing tooth enamel was obtained through combining a micro-sampling approach with proteomic analyses of erupted and unerupted permanent pig molars [15].

As expected, the amount of protein was seen to be higher during secretory stage and decreased from early enamel development to later stages and mature enamel. This pattern is paralleled in the abundance of amelogenin, ameloblastin, enamelin and Mmp20, whereas Klk4 was seen in maturing enamel. Several members of the serpin family were identified in this study with different distribution during different developmental stages of enamel, including antithrombin-III and serpin H1 (Figure 4). Different collagen types were also identified during secretory and maturation stage enamel. The presence of collagens is understood as a product of odontoblast secretion that can pass from the dentin into the enamel. Previous studies showed that collagen in enamel is not completely absorbed in the course of the mineralization and can interact with enamelin and amelogenin [124,125,126,127]. Albumin was also identified in this study throughout all stages of enamel development. Both studies of permanent porcine molar enamel showed similar patterns of protein removal with an increase in mineral hardening [15,112]. The differences between studies are certainly in part due to protocol and technique variations—Yamakoshi et al. used a two-dimensional protein fractionation system to analyze the enamel composition; Green at al. used microsamples of enamel subjected to trichloroacetic acid protein extraction, trypsin digestion and nano LC-MS/MS analysis [15,112]. Although both studies show a similar pattern of protein removal, why these nonspecific enamel proteins are present in the enamel matrix and what their role is in the hardening process is still unknown.

Taken together, these studies illustrate that the porcine model has been especially advantageous to study the spatial changes within the mineralizing tooth crown and throughout the thickness of the enamel layer. The similarity between human and porcine teeth is especially helpful for model systems of soft human enamel, when rate as well as depth of de- and remineralization processes into deeper enamel layers are of concern or when no genetically tractable mouse model exists for a human enamel defect.

## 9. Chalky Teeth and Serum Albumin in Enamel

An enamel malformation, commonly known as chalky teeth, is a dental developmental defect with unresolved etiology. Demarcated discoloration is characteristic for the affected enamel that is soft, hypomineralized and seen in first permanent molars at the time of their eruption, hence termed molar hypomineralization (MH) or referred to as molar-incisor hypomineralization (MIH), as incisors can also be affected. The staggering prevalence is globally 15% but 30% in the US [128,129]. Although the etiology of MH is still unknown, it is clear that an excess of retained organic matter compromises enamel mineralization [92,130,131,132,133,134]. The affected children experience severe pain and sensitivity and MH teeth are prone to caries, abrasion and fracture requiring surface coverage with fillings or replacement of the entire tooth crown or tooth extraction. Poor bonding of fillings and orthodontic needs after tooth extractions require repeated dental treatments, often performed under general anesthesia. Topical fluoride treatment is used clinically to promote enamel hardening. Yet, the overload of soft organic content embeds enamel crystallites and prevents their growth, thus also limiting the treatment [135].

The elevated protein content in MH enamel is not due to retained amelogenin and in this aspect distinct from other dental defects, such as AI where amelogenin residuals are high (Figure 3) [50,86,90,91,92,136]. The increasing severity of MH is classified by discoloration of the affected enamel from opaque/white in least defective, to yellow and brown in most affected enamel [48]. Interestingly, MH enamel is characterized by integrity of the enamel surface and high content of serum albumin, hemoglobin, alpha-1-antitrypsin, antithrombin III and serpin B3 (Figure 3) [50]. Ameloblastin was only identified in areas of brown discolored MH enamel, while amelogenin was not detected [48]. Serum albumin, type I collagen and alpha-1-antitrypsin were detected in all of the diagnostic MH discolorations, while antithrombin III was detected in yellow and brown demarcations. Both, antithrombin-III and alpha-1-antitrypsin are classified as serine protease inhibitors or proteins that inhibit the biosynthesis or activity of proteases such as Klk4 [137]. The relationship between serum albumin, Klk4 and the presence and activity of different protease inhibitors, such as antithrombin-III and alpha-1-antitrypsin, in MH teeth needs to be further explored. The accumulation of organic components in MH, AI and fluorosed teeth has been discussed extensively [48,50,88,89,90,91,93].

The detection of serum albumin in the protein matrix of healthy enamel has been controversial and characterized as contamination in many studies. However, the consistently reported identification of albumin, irrespective of tooth type, developmental stage of enamel and through the application of different analytical techniques, could also suggest that the protein is not an artifactual contamination. Albumin could be an exogenous protein that enters naturally and in vivo into porous tooth enamel, rather than being introduced through the handling and analytical procedures (Figure 2). Furthermore, albumin was described as component of the non-amelogenin fraction in enamel and has been detected during all the stages of enamel development with degradation products appearing during early maturation [138,139,140]. While albumin has been detected in healthy mature enamel at low concentrations, its presence in several defects is higher. This can be taken as an indication of its natural presence in the enamel and a compromised removal process in enamel defects [48,50,141]. Although the role of albumin in enamel development is unknown, it has been shown that albumin binds to apatite crystals and inhibits their growth in enamel [142,143,144]. These findings and the presence of albumin in secretory stage enamel have shaped the consideration that amelogenin could regulate crystal growth by preventing the binding of albumin to crystallite surfaces. This perspective is supported by evidence of proteolytic degradation of both amelogenin and albumin during maturation stage to allow for crystal growth [145]. The specific source of serum albumin and the mechanism leading to its presence in both sound and unsound enamel await to be resolved.

## 10. Proteomic Analysis in Enamel Cells

In 2001, the online proteomic database ToothPrint was created, providing a bioinformatic resource focusing on rat tissues, especially cells of tooth structures [146]. The information on ToothPrint includes proteins identified in ameloblasts and the enamel matrix based on data published between 1996 and 2002, with the last update recorded in 2010 [147,148,149,150].

During the maturation stage of enamel formation crystallites grow in thickness to form densely packed crystal bundles to achieve the final enamel hardness. This growth of calcium phosphate crystallites requires a massive amount of calcium ions. As calcium is quite toxic to cells, the question arises by which mechanisms ameloblasts transport such large amounts of ions to the extracellular matrix filling the confined space between the dentin and the ameloblast layer. Some studies confirmed a high abundance of calcium-binding proteins in the developing enamel of rats to support the process of calcium transport [147,148,149,150]. Calbidin28kDa and calmodulin were identified as high-affinity calcium-binding proteins [148]. However, Calbidin28kDa expression is higher in secretory ameloblast than during the maturation phase, when most mineral is formed. This suggests that other proteins are involved in the transcellular passage of calcium. However, no other high-affinity protein was found up-regulated during enamel maturation, instead two low-affinity calcium-binding proteins were found, Calreticulin and Endoplasmin [149]. Both proteins are endoplasmic reticulum members, providing evidence for calcium transport through the endoplasmic reticulum. The mitochondrial ATP synthase F1-β-subunit was identified by MALDI-TOF-MS [150]. This protein was characterized as having low affinity and moderate capacity for calcium-binding but found to be abundant and up regulated during maturation stage. Taken together, these findings are excellent examples of proteomic approaches advancing our understanding of cellular mechanisms in enamel formation.

In addition to global protein analysis, proteomics has been used successfully to localize specific proteins or pathways in cells of the enamel epithelium [147]. Mangum et al. applied different proteomic approaches to characterize the ameloblast proteome from rat pups, thus increasing the number and type of proteins identified compared with previous analyses [147,151,152]. The enamel epithelia were divided into three fractions and each of them was processed differently according to the sample characteristics. The cytosolic fraction was subjected to 2D-PAGE followed by MALDI-TOF/TOF spot analysis. The cytoskeletal and nuclear fractions were analyzed by sodium dodecyl sulfate polyacrylamide gel electrophoresis (SDS-PAGE) followed by LC-MS/MS (Gel-LC-MS/MS). Some proteins that are invisible to 2D-PAGE due to their large size or high charge were identified by Gel-LC-MS/MS. While this study highlights advances in proteomic analyses regarding sensitivity and identification, the findings also underscore the importance of choosing the proteomics approach that is most appropriate for the samples and question at hand. The gel-free methods used in enamel proteomics studies, MALDI-TOF/TOF and Nano LC-MS/MS, provide high sensitivity and accuracy [153]. Whereas MALDI-TOF/TOF data acquisition is fast, Nano LC-MS/MS offers the two important advantages of higher sensitivity and being more reliable. Continued advances in instrument sensitivity and refinement of extraction methods will no doubt lead to more peptides detected in fully mineralized enamel and expand our understanding of both enamel formation and the information entombed in enamel.

## 11. Conclusions

Tooth enamel forms in an ephemeral proteinaceous matrix that is not fully characterized for its composition or the localization and functionality of all its components during enamel development. Proteomic studies, however, have been critical to expand the list of known enamel components and to advance our understanding of changes in enamel composition during formation of healthy and defective enamel. The comparison between earlier and recent publications highlights how technical advances in mass spectrometry such as gel-free methods MALDI-TOF/TOF and Nano LC-MS/MS have increased both sensitivity and identification in proteomic analyses, yet also complicate the comparison of findings. To fully leverage a synthesis of these findings, standard protocols are urgently needed for the extraction, manipulation and storage of teeth, as well as for protein extraction, especially because the low availability of human samples and low protein content in enamel pose limitations to data acquisition and analyses. The combination of “omics” to include proteomics, genomics, transcriptomics and metabolomics will be key to better understand enamel mineralization and the underlying mechanisms of enamel defects.

Equally important will be what we learn from animal models about tooth enamel mineralization (Table 3). The majority of the works discussed in this review is focused on enamel protein identification. The global characterization of protein composition is fundamental to understand the functions, processes and pathways involved in this mineralizing matrix. While not well characterized for all structural enamel matrix proteins during the course of enamel formation, it is clear that their post-translational modifications are critical for proper enamel mineralization. However, the mechanisms and pathways pertaining to post-translational modifications remain unresolved (Figure 5) and point to future directions for proteomics analyses of tooth enamel formation. For example, the use of cell cultures would allow experimental control of growing conditions and study their effects on protein secretion and post-translational modifications. Complementing classical histology approaches, proteomic analysis offer a discovery driven approach to uncover new players in enamel mineralization, discern mechanisms of secretion through classical or non-classical pathways and provide new characterization and functional insights on post-translational modifications.

## Figures and Tables

**Figure 1 ijms-21-04458-f001:**
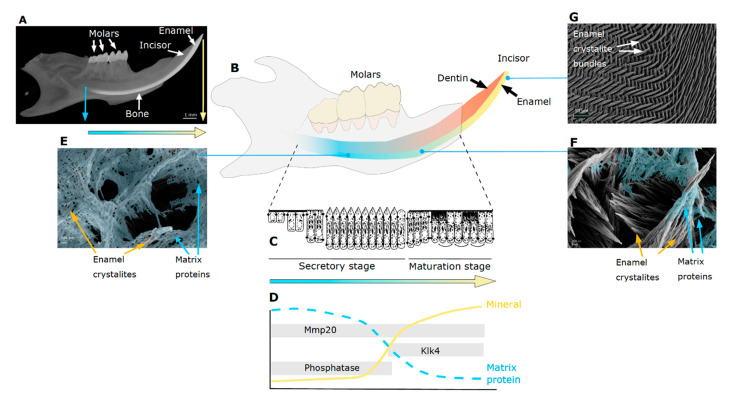
Illustration of the process of enamel formation. (**A**) Micro computed tomography radiograph of mouse mandible (without mandibular ramus) showing lighter grey values for higher mineral density, tooth enamel in white on molar crowns and in a gradient according to increasing mineralization in the continuously forming incisor. Data acquired on a Scanco Medical μ40 instrument (Scanco, Brüttisellen, CH). (**B**) Schematic representation of mouse mandible showing continuous incisor growth from early stages of enamel formation with high amount of protein (blue) to mature, erupted enamel containing less than 1% by weight protein and more than 95 % by weight mineral (yellow). (**C**) Schematic of ameloblast morphology indicating differentiation and different stages during enamel formation (adapted from Hu et al. [11]). Color filled arrow indicates progression of time and compositional changes of the mineralizing matrix from high protein content (blue) with little mineral, to low protein and high mineral content (yellow) in later stages of mineralization and mature enamel. (**D**) Graph depicting decreasing protein and increasing mineral content in developing enamel, with activity period of matrix metalloproteinase 20 (Mmp20), phosphatase and kallikrein-4 (Klk4) during mineralization of rodent incisors (adapted from Robinson et al. [12]). (**E**–**G**) Scanning electron microscopy (SEM) images of the forming mouse incisor in longitudinal plane. Enamel with high protein content seen at high magnification in late secretory stage (**E**, scale bar 200 nm) and with some matrix protein, colored in blue, around crystal bundles in early maturation stage (**F**, scale bar 200 nm). At lower magnification, the decussation pattern of crystal bundles in mature, erupted enamel in (**G**, scale bar 10 microns). SEM—samples were fixed, dehydrated, epoxy resin embedded, polished, phosphoric acid etched, gold coated and imaged on a Zeiss SEM Ultra55.

**Figure 2 ijms-21-04458-f002:**
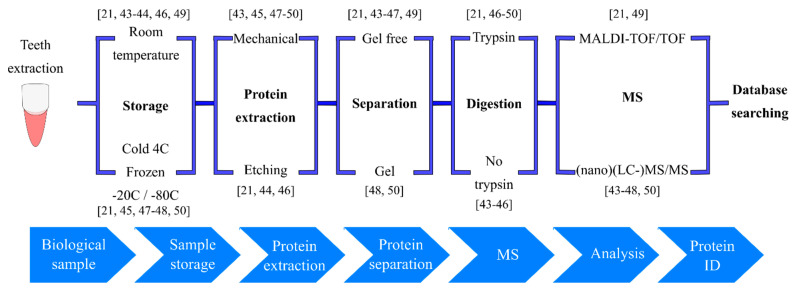
Strategies for proteomic analyses of tooth enamel. Schematic of treatment steps used to characterize proteins in human tooth enamel from archeological, modern and teeth affected by molar hypomineralization (chalky tooth enamel). Porto et al. [21]; Lugli et al. [43]; Stewart et al. [44]; Castiblanco et al. [45]; Stewart et al. [46]; Jagr et al. [47]; Farah et al. [48]; Nielsen-Marsh et al. [49]; Mangum et al. [50]. Each bracket denotes alternative options and reflects the variability in published protocols.

**Figure 3 ijms-21-04458-f003:**
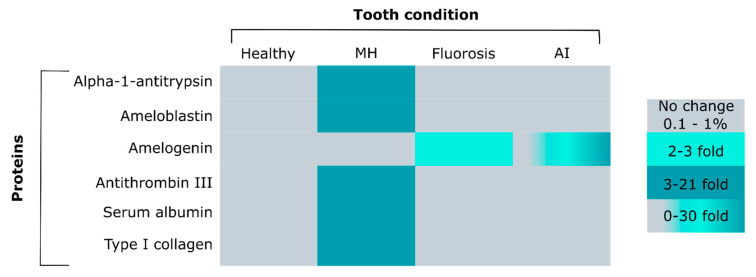
Comparison of protein content between healthy and diseased tooth enamel. Shown proteins are seen in (rows) identified in any of the tooth conditions (columns) are illustrated. Healthy teeth shown as reference in light grey, Chalky/ Molar Hypomolarization (MH): enamel affected by molar hypomineralization [48,50,92], Fluorosis [93,94,95] and Amelogenesis Imperfecta (AI): hypocalcified and hypomaturation amelogenesis imperfecta enamel [88,89,90]; Range of percent by weight (wt%) of protein abundance relative to healthy enamel show in colors: light gray for healthy range of 0.1–1 wt%; 2–3 fold increase (light teal); 3–30 fold increase (dark teal); 0–30 fold increase (grey-teal gradient).

**Figure 4 ijms-21-04458-f004:**
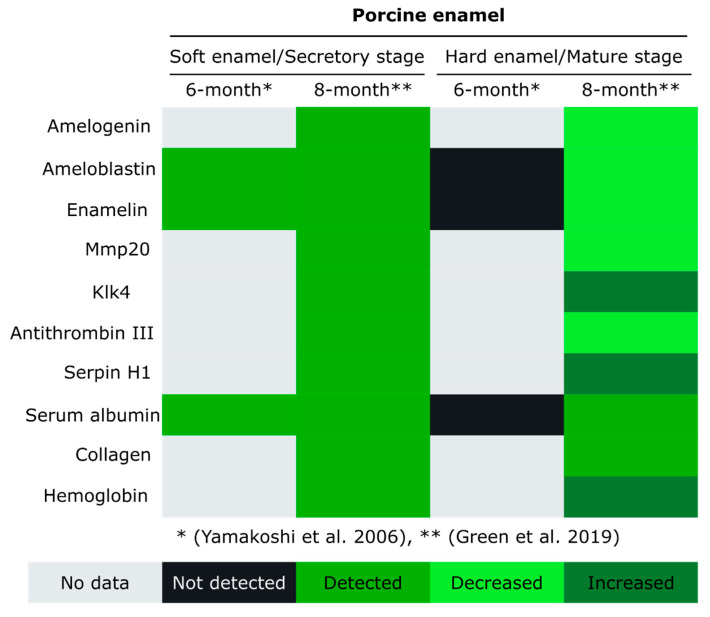
Overview of proteins detected in enamel from permanent molars of 6-months-old [112] and 8-months-old pigs [15]. The chart distribution is based on the protein detection in soft enamel and hard enamel, described as secretory and mature stages, respectively.

**Figure 5 ijms-21-04458-f005:**
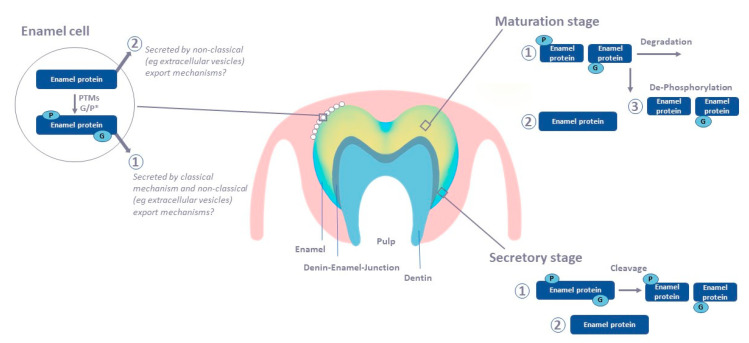
Schematic of potential mechanism of post-translational modifications and secretion of enamel proteins during the course of enamel formation. (1) Post-translational modifications of enamel proteins occur with the ameloblast. These modifications are used as a signal to be secreted, cleavage during the secretory stage and degraded during the maturation stage. (2) Some of the enamel proteins generated with the ameloblast lack post-translational modification and they are secreted via non-classical pathways. These proteins would be less or not degraded during the mineralization process. (3) Alternative to pathway 1, de-phosphorylation to account for the presence of non-phosphorylated enamel proteins in mature enamel. (P) phosphorylation, (G) glycosylation; Blue: enamel that is more recently deposited, younger, with higher protein content. Yellow: enamel that is further developed and more mature, older enamel, more mineralized with lower protein content.

**Table 1 ijms-21-04458-t001:** **Peptide sequences identified from amelogenin.** Only peptides identified in a minimum of 2 samples are illustrated in this table. Peptides from isoform Y are annotated. Porto et al. [21]; Lugli et al. [43]; Stewart et al. [44]; Castiblanco et al. [45]; Stewart et al. [46].

Peptide(s)	Contemporary Molar (Trypsin)	Contemporary Molar (No Trypsin)	Mummy (Trypsin)	Mummy (No Trypsin)
IRPPYPSYGYEPMG		[45,46]		
LPPHPGHPGYIN	[21,46]	[21]		
LPPHPGHPGYINF	[46]	[45,46]		
LPPHPGHPGYINFSYEVLTPLK	[46]	[45]		
M(ox)PLPPHPGH (AMELX/AMELY)		[43,46]		[43,46]
M(ox)PLPPHPGHPGYINF	[46]	[43,46]		[43]
MPLPPHPGHPG	[21,46]	[45,46]		
MPLPPHPGHPGYIN	[46]	[45,46]		
MPLPPHPGHPGYINF		[45,46]	[21]	
MPLPPHPGHPGYINFSYEVLTPLK	[46]	[45]		
PHPGHPGYINF	[46]	[45,46]		[46]
SIRPPYPSY	[46]	[43,46]		[43,44,46]
SIRPPYPSYGYEP		[45,46]		
SIRPPYPSYGYEPM		[45,46]		
SIRPPYPSYGYEPMG		[43,45,46]		[43]
SM(ox)IRPPY (AMELY)		[43,46]		[43,44]
SYEVLTPLK (AMELX/AMELY)	[46]	[43,45,46]		[43,46]
SYEVLTPLKWYQSIRPPYP	[46]	[43]		[43]
WYQSIRPPYP	[46]		[21]	
YEVLTPLK	[46]	[45,46]		[46]
YEVLTPLKWY (AMELX/AMELY)		[43,46]		[43,46]

**Table 2 ijms-21-04458-t002:** Proteins identified from contemporary and archeological enamel samples.

Condition	Proteins (Peptides Identified)											
	Amelogenin, X Isoform	Amelogenin, Y Isoform	Ameloblastin	Enamelin	Serum Albumin	Hemoglobin Subunit Alpha	Hemoglobin Subunit Beta	Collagen Alpha-1(I) Chain	Collagen Alpha-1(III) Chain	Collagen Alpha-2(I) Chain	Antithrombin-III	Alpha-1-Antitrypsin
Contemporary (trypsin)	[21,46]	[21,46]	[46,47,48]	[46]	[46,47,48]	[46,47]	[46,47]	[46,47,48]	[47]	[46,47,48]	[47]	[46,47,48]
Contemporary (no trypsin)	[43,44,45,46]	[43,44,45,46]	[43,44,45,46]	[43,44,45,46]	[45,46]	[46]	[46]			[43]	[45]	
Mummy (no trypsin)	[43,46]	[43,46]	[43,46]	[43,46]						[43,46]		
Mummy (trypsin)	[21]											

The use of trypsin digestion after protein extraction and before mass spectrometry is specified in the table. Porto et al. [21]; Lugli et al. [43]; Castiblanco et al. [45]; Stewart et al. [46]; Jagr et al. [47]; Farah et al. [48].

**Table 3 ijms-21-04458-t003:** Summary of pros and cons of the species for research on tooth enamel. Per model, the most prominent pros and cons on the models covered in this review are listed.

Model	References	Pros	Cons
Human: modern, archeological 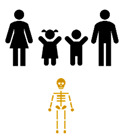	[2,3,21,34,37,40,41,43,44,45,46,47,48,49,50,51,52,54,86,88,89,90,91,92,93,95,99,100,106]	Primary source for research on human enamel formation Clinical relevance of factors affecting enamel development Contemporary and archaeological record of ontogeny, health history, behavior and environment Forensic sciences: sex determination in human remains	Availability limited (shed teeth, teeth extracted for clinical reasons or postmortem) Limited access to forming teeth Small amount of protein in erupted teeth No deliberate genetic models to study genotype effects on enamel Difficulty to relate genetic and epigenetic variability to subtle enamel variability No mechanistic studies, no controlled experiments
Pig/miniature pig 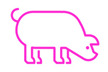	[15,35,36,71,112,121,122]	Availability Large size provides big sample amount Similar to human teeth in size: good model system Access to forming teeth, matrix composition in different stages of enamel development Can study post-transcriptional modifications of matrix proteins (phosphorylation, glycosylation) Mechanistic studies and controlled experiments possible	Gene sequence and processing vary from human, e.g., amelogenin with 173 amino acids vs 189 in humans) Poorly annotated genome and proteome Controlled studies: costs for animal housing Genetic modification possible but limited
Mouse/rat 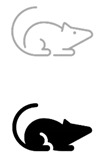	[37,56,57,58,75,94,102,105,107,108,109,111,115,117,118,119,120]	Availability Low cost Dental development well characterized Continuously growing incisors allow access to all stages for enamel formation Can study enamel matrix in different stages of development Genetically tractable: can explain variability in phenotype and properties Genetic modification: mechanistic studies possible Mechanistic studies and controlled experiments most feasible compared to other mammals Disease models can be created or are available.	Differences in number and morphology of teeth Monophyodont dentition (one set of teeth, not primary and permanent teeth as in humans and pigs) Small tooth size: low amount of sample material to analyze and difficulty of physically isolating enamel matrix. Sample pooling from many animals analyze enamel only at specific stages of formation Gene sequence and processing vary from human, e.g., amelogenin with 180 amino acids vs. 189 in humans) Amelogenin encoded only by one gene (AMELX).

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
