# Peer review of "Tooth Enamel and Its Dynamic Protein Matrix"

_ijms, 2020, doi:10.3390/ijms21124458_

Round 1

Reviewer 1 Report

This review summarizes the current knowledge and published proteomics data of healthy and diseased tooth enamel, including advancements in forensic applications and disease models in animals. The topic was covered in excellent detail and referenced extremely well.

In Table 1, Table 2, and Figure 2, it would be better to just enter the reference instead of the number.

Author Response

Comment 1: In Table 1, Table 2, and Figure 2, it would be better to just enter the reference instead of the number.

Response 1: Thank you for your reading of the manuscript and providing constructive feedback. We appreciate the suggestion. The numbers from the Tables 1 and 2, and Figure 2 have been removed and the reference numbers have been added. This change certainly improves the manuscript and will makes the paper easier to follow.

These changes are in lines 152 to 165 of the manuscript with tracked changes.

Reviewer 2 Report

The present review entitled “Tooth enamel and its dynamic protein matrix” is an interesting work not only for the researcher who is working in dentistry but also for hard tissue related issues for example bone. Authors highlighted the relationship between tooth enamel and protein matrix. A review in the area would be of great interest to the common readership of the journal.

Following suggestions could improve the manuscript:

  1. I suggest the authors may include a table summarizing the studies covered on animal and human model along with the pros and cons of the Tooth enamel.
  2. The presentation of the Manuscript is very well. But still I believe that the manuscript lacking with coherence and in-depth molecular mechanism. Maybe, a diagrammatic representation (an overview of the study) will be useful to add in the manuscript.
  3. Authors should provide detail paragraph on the applications and future perspectives specially in enamel organ cell culture and tissue engineering, which is missing in this review.
  4. Some sentences in the manuscript are quite long. For clear meaning, I will suggest to rephrase or break into 2 sentences. English and grammar level in the manuscript is good enough for publication.

Author Response

We thank the reviewer for his very helpful advice and suggestions.

Comment 1: I suggest the authors may include a table summarizing the studies covered on animal and human model along with the pros and cons of the Tooth enamel.

Response 1: We appreciate this good idea and a new table has been included, Table 3, at the end of the paper (lines 520 and 533-536 of the manuscript with tracked changes). This new table includes the pros and cons of the tooth enamel research covered on human and animal models. This table summarizes the content of the review, making it clearer. 

Comment 2: The presentation of the Manuscript is very well. But still I believe that the manuscript lacking with coherence and in-depth molecular mechanism. Maybe, a diagrammatic representation (an overview of the study) will be useful to add in the manuscript.

Response 2: There is a surprising number of unresolved questions regarding the formation of tooth enamel, from a full characterization of the composition of the mineralizing enamel matrix to the uncovering of mechanisms underlying enamel mineralization. Accordingly, proteomic analyses in discovery mode are still contributing to the inventory and our understanding of enamel matrix composition. To address the perceived lack with consistency and in-depth molecular mechanism, we have focused on a mechanism in a newly added section (lines 468-506, in response to Comment 3). In addition, we added paragraph about application and future perspectives to the conclusions section (lines 521-532 of the manuscript with tracked changes), and a new figure, Figure 5 (lines 537-546 of the manuscript with tracked changes). The paragraph is focused on the unresolved questions regarding post-translational modifications and secretion of enamel proteins. We have also included a suggestion of future directions. 

Comment 3: Authors should provide detail paragraph on the applications and future perspectives specially in enamel organ cell culture and tissue engineering, which is missing in this review.

Response 3: A new section about enamel organ cell culture has been added: “Section 10. Proteomic analysis in enamel cells”, Lines 468-506 of the manuscript with tracked changes. This new section presents the studies carried about cellular mechanisms in enamel formation using proteomics to study. With the information added, this review covers all the information regarding the proteomic studies published in enamel and improves the content and quality of the review.

Comment 4: Some sentences in the manuscript are quite long. For clear meaning, I will suggest to rephrase or break into 2 sentences. English and grammar level in the manuscript is good enough for publication.

Response 4: Thank you for this feedback. These sentences have been changed in the new version of the manuscript and make the writing much more accessible.

Lines 73-78, 105-108, 129-133, 133-136, 168-170, 175-178, 263-268, 279-283, 284-287, 297-300, 328-330, 340-342, 376-379, 388-390, 411-415 and 433-435 of the manuscript with tracked changes.